# Comparison of Illumina versus Nanopore 16S rRNA Gene Sequencing of the Human Nasal Microbiota

**DOI:** 10.3390/genes11091105

**Published:** 2020-09-21

**Authors:** Astrid P. Heikema, Deborah Horst-Kreft, Stefan A. Boers, Rick Jansen, Saskia D. Hiltemann, Willem de Koning, Robert Kraaij, Maria A. J. de Ridder, Chantal B. van Houten, Louis J. Bont, Andrew P. Stubbs, John P. Hays

**Affiliations:** 1Department of Medical Microbiology and Infectious Diseases, Erasmus University Medical Center (Erasmus MC), 3015 CN Rotterdam, The Netherlands; d.kreft@erasmusmc.nl (D.H.-K.); j.hays@erasmusmc.nl (J.P.H.); 2Department of Microbiology, Leiden University Medical Center (LUMC), 2333 ZA Leiden, The Netherlands; s.a.boers@lumc.nl; 3Department of Pathology, Erasmus University Medical Center (Erasmus MC), 3015 CN Rotterdam, The Netherlands; r.jansen.1@erasmusmc.nl (R.J.); s.hiltemann@erasmusmc.nl (S.D.H.); w.dekoning.1@erasmusmc.nl (W.d.K.); a.stubbs@erasmusmc.nl (A.P.S.); 4Department of Internal Medicine, Erasmus University Medical Center (Erasmus MC), 3015 CN Rotterdam, The Netherlands; r.kraaij@erasmusmc.nl; 5Department of Medical Informatics, Erasmus University Medical Center (Erasmus MC), 3015 CN Rotterdam, The Netherlands; m.deridder@erasmusmc.nl; 6Division of Paediatric Immunology and Infectious Diseases, University Medical Center Utrecht, 3584 CX Utrecht, The Netherlands; c.b.vanhouten@umcutrecht.nl (C.B.v.H.); l.bont@umcutrecht.nl (L.J.B.)

**Keywords:** nasal microbiota, Illumina sequencing, nanopore sequencing, 16S rRNA gene, bacterial species, *Corynebacterium*

## Abstract

Illumina and nanopore sequencing technologies are powerful tools that can be used to determine the bacterial composition of complex microbial communities. In this study, we compared nasal microbiota results at genus level using both Illumina and nanopore 16S rRNA gene sequencing. We also monitored the progression of nanopore sequencing in the accurate identification of species, using pure, single species cultures, and evaluated the performance of the nanopore EPI2ME 16S data analysis pipeline. Fifty-nine nasal swabs were sequenced using Illumina MiSeq and Oxford Nanopore 16S rRNA gene sequencing technologies. In addition, five pure cultures of relevant bacterial species were sequenced with the nanopore sequencing technology. The Illumina MiSeq sequence data were processed using bioinformatics modules present in the Mothur software package. Albacore and Guppy base calling, a workflow in nanopore EPI2ME (Oxford Nanopore Technologies—ONT, Oxford, UK) and an in-house developed bioinformatics script were used to analyze the nanopore data. At genus level, similar bacterial diversity profiles were found, and five main and established genera were identified by both platforms. However, probably due to mismatching of the nanopore sequence primers, the nanopore sequencing platform identified *Corynebacterium* in much lower abundance compared to Illumina sequencing. Further, when using default settings in the EPI2ME workflow, almost all sequence reads that seem to belong to the bacterial genus *Dolosigranulum* and a considerable part to the genus *Haemophilus* were only identified at family level. Nanopore sequencing of single species cultures demonstrated at least 88% accurate identification of the species at genus and species level for 4/5 strains tested, including improvements in accurate sequence read identification when the basecaller Guppy and Albacore, and when flowcell versions R9.4 (Oxford Nanopore Technologies—ONT, Oxford, UK) and R9.2 (Oxford Nanopore Technologies—ONT, Oxford, UK) were compared. In conclusion, the current study shows that the nanopore sequencing platform is comparable with the Illumina platform in detection bacterial genera of the nasal microbiota, but the nanopore platform does have problems in detecting bacteria within the genus *Corynebacterium*. Although advances are being made, thorough validation of the nanopore platform is still recommendable.

## 1. Introduction

The use of traditional culture and established 16S rRNA gene sequencing techniques has shown that the composition of the nasal microbiota comprises microbiota profiles, dominated by four or five microbial genera. The microbiota composition varies in individuals with age [1], and shows large-scale variations in the first few years of life [2]. This variation usually involves colonization with *Streptococcus pneumoniae*, *Haemophilus influenzae* and *Moraxella catarrhalis* (three bacterial species often associated with the development of upper respiratory tract infections, including otitis media in young children) as well as *Staphylococcus aureus*, *Dolosigranulum* sp. or *Corynebacterium* spp. Further, the composition of the nasal microbiota has been associated with several other diseases, including the progression of cystic fibrosis [3], chronic rhinosinusitis [4], and progression to pneumonia after respiratory syncytial virus upper respiratory tract infection [5]. Nasal colonization with bacterial species such as *Streptococcus pneumoniae*, *Haemophilus influenza*, *Moraxella catarrhalis,* and *Staphylococcus aureus* may in the majority of cases be mutualistic or commensal, though a disturbance in this symbiotic relationship could lead to dysbiosis and disease, especially when these bacteria may also be present in the nasopharynx [6]. However, this phenomenon may not be related to microbiota profiles alone, but to a combination of bacterial, viral and child characteristics [7].

Unfortunately, traditional culture techniques are unable to detect a wide range of the so-called ‘non-culturable’ bacteria that DNA sequencing techniques have indicated to be present within the human nasal microbiota [8]. Also, to date, accurate species identification using 16S rRNA gene sequencing protocols in combination with the most popular sequencing platform (Illumina sequencing) is currently not universally possible as only short regions of bacterial 16S rRNA genes tend to be sequenced using Illumina technology [9]. This means that the majority of microbiota publications to date have been limited to reporting the diversity of the (nasal) microbiota at best at the genus level. However, the accurate speciation of bacteria can be very important for clinicians as a bacterial genus may contain several species that possess very different virulence characteristics [10]. For example, being able to differentiate between a *Staphylococcus aureus* and a *Staphylococcus epidermidis* infection may be significant in the treatment of sepsis or skin infections.

Nanopore sequencing (Oxford Nanopore Technologies—ONT, Oxford, UK) [11], is a ‘third generation’ (i.e., single-molecule) sequencing technology that is able to generate long sequence read-lengths that can span the majority of the bacterial 16S rRNA gene. Several recent comparative studies demonstrated promising results for the nanopore technology including identification of the microbiota composition at the species level. For example, a significantly similar bacterial composition at genus level and the identification of more bacterial species was reported when Oxford Nanopore and Illumina 16S rRNA gene sequencing were compared for the mouse gut microbiota [12]. In another study, the performance of nanopore versus IonTorrent PGM^®^ sequencing on mock and dog skin microbiota samples indicated increased bacterial richness at high taxonomic levels (species identification) associated with nanopore sequencing [13]. In a separate time course analysis, nanopore 16S rRNA gene sequencing resulted in the detection of all 20 of the bacterial species present in a mock bacterial community within minutes [14]. A drawback of nanopore sequencing is the relatively high sequencing error rate, ranging from 5% [1] to 38.2% [15]. This further complicates accurate taxonomy at species level, particularly for bacterial species with a high sequence similarity in the 16S RNA gene.

Although comparisons of nanopore sequencing with other sequencing systems have previously been published, to our knowledge no comparative data were published with a specific focus on the nasal microbiota. The nasal microbiota contains microbial species at lower microbial abundance compared to high-biomass samples such as feces. It, however, may be a source of potential antibiotic-resistant pathogens such as methicillin-resistant *Staphylococcus aureus* (MRSA) [16]. In this manuscript, we compared Illumina versus nanopore sequencing at genus level using nose swab samples that had been obtained from the European Union-funded FP7 project [17]. Initial comparative research was performed using version R9.2 nanopore sequencing devices (flowcells), the Albacore basecaller and earlier versions of the EPI2ME 16S sequence data analysis pipeline, which is still evolving and being updated by ONT [18]. Therefore, subsequent to, and based on, the results of our initial comparative analysis, we performed further analysis and investigated the potential effect of newer ONT advancements (EPI2ME, the Guppy basecaller and flowcells R9.4) on the results of microbiota profiling at genus and species level using pure cultures of relevant bacterial species.

## 2. Material and Methods

### 2.1. Sample Collection and Selection

Fifty-nine nose swab samples generating at least 1000 Illumina sequence reads and 3 × 10^3^ 16S rRNA gene copies per microliter were randomly selected for nanopore 16S rRNA gene sequencing. These samples had been previously obtained from patients with lower respiratory tract infections, sepsis, and non-infected control patients participating in the EU FP7-funded TAILORED-treatment study, and Illumina sequenced. They comprised nose swab samples from 10 adults and 49 children under the age of 18. Seven negative control swabs were also sequenced, containing nasal swab Universal Transport Medium (UTM, ESwab™, COPAN Diagnostics Inc., Brescia, Italy) only.

### 2.2. DNA Isolation

DNA was previously isolated from nasal swab samples using the mag mini kit (LGC Standards, Wesel, Germany) and an adjusted protocol that included an initial bead-beating step. In short, 200 µL of nose swab medium combined with 200 µL phenol and 150 µL Lysis buffer BL (LGC Standards, Wesel, Germany) was added to a vial containing Lysing Matrix beads (MP Biomedicals, Eschwege, Germany) and subjected to bead-beating using a FastPrep-24 (MP Biomedicals, Eschwege, Germany) at 6m/s for 60 s. After centrifugation, 200 µL of the water phase (top layer) was incubated for 2 min at room temperature with 400 µL binding buffer BL (LGC Standards, Wesel, Germany), to which 10 µL mag particle suspension (LGC Standards, Wesel, Germany) had been added. The manufacturer’s protocol was then followed, with the exception that the DNA was eluted by incubating for 30 min at 55 °C instead of 10 min. Prior to 16S rRNA gene sequencing, the total number of 16S rRNA gene copy numbers within each DNA extract was measured using a 16S rRNA gene quantitative PCR as previously described [19].

### 2.3. Bacterial Strains

The following purely cultured bacterial strains were used in this study: *Haemophilus influenzea* ATCC 10211, *Moraxella catarrhalis* ATCC 25240, *Staphylococcus aureus* ATCC 25923, *Staphylococcus epidermidis* ATCC 12228, *Streptococcus pneumoniae* ATCC 49619, *Corynebacterium diphtheria* ATCC 13812, and from our own hospital strain collection: *Corynebacterium accollens*, *Corynebacterium amycolatum*, *Corynebacterium pseudodiphtheriticum*, and *Corynebacterium striatum*. The identity of the hospital isolates used was confirmed by matrix-assisted laser desorption ionization-time of flight spectrometry (MALDI-TOF MS, Bruker Daltonics).

### 2.4. Illumina Sequencing

The hypervariable V5 and V6 regions (276 base pairs—bp) of the 16S rRNA gene were amplified using the 785F (5′-GGA TTA GAT ACC CBR GTA GTC-3′) and 1061R (5′-TCA CGR CAC GAG CTG ACG AC-3′) primers [20], and dual indexing [21]. Amplicons were generated in 30 cycli using the FastStart High Fidelity System (Roche, Woerden, The Netherlands), normalized using the SequalPrep Normalization Plate kit (Thermo Fischer Scientific, Breda, The Netherlands) and pooled in batches of approximately 250 samples. Pools were purified prior to sequencing using the Agencourt AMPure XP (Beckman Coulter Life Science, Indianapolis, IN, USA), and the amplicon size and quantity of the pools were assessed on the LabChip GX (PerkinElmer Inc., Groningen, The Netherlands). The PhiX Control v3 library (Illumina Inc., San Diego, CA, USA) was combined (~10%) with the pooled amplicon libraries and each pool was sequenced on an Illumina MiSeq sequencer (MiSeq Reagent Kit v3, 2 × 300 bp).

### 2.5. Nanopore Sequencing

16S rRNA gene sequence libraries were prepared with the 16S Rapid Amplicon Barcoding Kit (Oxford Nanopore Technologies—ONT, Oxford, UK, SQK-RAB201) according to the standard procedures described by ONT. The complete 16S rRNA gene was amplified using 10 µL input DNA purified from nasal swabs, LongAmp^®^ Taq 2× master mix (New England Biolabs, Ipswich, MA, USA) and the barcoded nanopore sequence primers 27F 5′-AGA GTT TGA TCM TGG CTC AG-3′ and 149R 5′-CGG TTA CCT TGT TAC GAC TT-3′. The DNA amplification was performed on a T100 Thermal Cycler (Biorad, Lunteren, The Netherlands) using the program; 1 min denaturation at 95 °C, 25 cycles (95 °C—20 s, 55 °C—30 s, 68 °C—2 min) and a final extension step of 5 min at 65 °C. The 16S rRNA gene amplicons were quantified using Quant-IT™ PicoGreen™ (Thermo Fisher Scientific, Breda, The Netherlands), equal amounts of amplicons per sample were pooled and the library was further processed as described by the manufacturer. Next, the library was incubated with Library Loading Beads (Oxford Nanopore Technologies—ONT, Oxford, UK) and the mixture was added to the MinIon/GridIon flow cell (version R9.2 or R.9.4, Oxford Nanopore Technologies—ONT, Oxford, UK). Sequencing was performed using a MinIon or GridIon nanopore sequencer (Oxford Nanopore Technologies—ONT, Oxford, UK) for approximately 16 h.

### 2.6. Data Analysis

The Illumina MiSeq sequence data were analyzed using bioinformatics modules present in the Mothur software package [22], that we previously integrated into Galaxy (i.e., Galaxy mothur Toolset, Gm [23]). In short, forward and reverse FASTQ-formatted sequence files were merged using the make.contigs command. Primer sequences were trimmed and sequences that had an ambiguous base call (N) in the sequence or with lengths smaller than 200 were removed from the analysis. Unique sequences were then aligned against a customized reference alignment based on the SILVA reference alignment release 123 [24,25]. The reference sequences were trimmed to only include the V5–V6 region of the 16S rRNA gene using the pcr.seqs command. Sequences that did not align to this region were culled from further analysis and the alignments were trimmed so that the sequences fully overlapped the same alignment coordinates. Next, sequences were further de-noised by pre-clustering the sequences using the pre.cluster command allowing for up to two differences between sequences, and potentially chimeric sequences were removed using Uchime, as implemented in Mothur. The remaining sequences were classified using the classify.seqs command with the customized SILVA alignment release 123 as reference. Finally, sequences were clustered into operational taxonomic units (OTUs) at 97% similarity using the default settings of the dist.seq and cluster commands respectively, and the classify.otu algorithm was used to get a consensus taxonomy for each OTU.

Basecalling of nanopore signals was performed using the MinKNOW (MinION software, version 1.6, Oxford Nanopore Technologies—ONT, Oxford, UK) embedded Albacore version 1.0 data processing pipeline or the Guppy version 3.2.10 pipeline (Oxford Nanopore Technologies—ONT, Oxford, UK). The Barcoding workflow in the Metrichor Ltd. analysis platform EPI2ME (Oxford Nanopore Technologies—ONT, Oxford, UK) [26] was used for the de-barcoding of the sequence reads derived from the nose swab samples sequenced with the Oxford Nanopore platform. For the identification of bacteria at genes and species level, fast5 or fastq files containing full length 16S rRNA gene amplicons where uploaded to the EPI2ME desktop agent 16S workflow (versions 2.47.53720F8, 2.48.690655 or 2020.2.10, Oxford Nanopore Technologies—ONT, Oxford, UK) where each file was classified real-time using the NCBI 16S rRNA gene blast database [27]. Blastn was run using the parameters max_target seqs = 3 (finds the top three hits that are statistically significant) and output fmt = 6. The number of genera represented in the top three classifications (num_genus_taxid) was calculated along with the genus rank (if classified at genus rank or below) per sequencing record. These were calculated using the Python library ete2 [28], which utilizes the NCBI taxonomy. The top scoring classification per individual record within the file was selected as the read classification along with the accompanying num_genus_taxid and genus and species information. Coverage information per read was calculated as number of identical matches/query length. All read classifications were then filtered for >77% accuracy and >30% coverage, which removes spurious alignments. Results were returned via a web report and can be downloaded as a comma-separated values (CSV) file.

Then, the results in the CSV file of the EPI2ME 16S workflow output were used for further analysis using an in-house-generated Python script together with the Python ete2 package. This script reads the contents of the CSV file and retrieves the species and genus names from the NCBI taxonomy IDs found by the EPI2ME 16S workflow. Exclusion criteria for the single nanopore reads were an alignment count accuracy <80%, quality score (QC) score <7, read length <1400 >1700 bp, and a num_genus_taxid other than 1 or 2. These exclusion criteria apply for the initial analyses of the nasal swab samples in this study. For the nasal swab samples that were re-basecalled with Guppy, and the purely cultured bacterial strains that were (re-) basecalled with Guppy, the applied exclusion criteria were: alignment count accuracy 85%, QC score <9, read length <1400 >1700 bp, and an lca score other than 0. For species level identification, similar criteria and the highest scoring BLAST identification (top rank) was used. The higher accuracy and QC thresholds were chosen because (re-) basecalling with Guppy or using a R.9.4 flowcell resulted in a higher average QC score (from at least 7 to ~10) and accuracy (from ~85% to ~90%) in the EP2ME analysis (R9.2 flowcell, Albacore basecalling versus R9.2 or R9.4 flowcell and Guppy basecalling, respectively, data not shown). On average, ~15% of the reads were excluded after re-basecalling with Guppy and filtering with the more stringent thresholds (data not shown).

### 2.7. Statistics

Rarefaction analysis was performed to determine the amount of reads needed to accurately assess the bacteria richness in the samples (Appendix A). Plots were generated with QIIME (Quantitative Insights Into Microbial Ecology) version 1.9.1 (multiple_rarefactions.py, alpha_diversity.py, collate_alpha.py, make_rarefaction_plots.py) using the Shannon diversity metric. Based on the rarefraction analysis, samples generating >500 sequence reads were included for bioinformatics analysis.

Taxonomy results of the data produced after Illumina and nanopore sequencing were loaded into BioNumerics software version 7.6 (Applied Math, Sint-Martens-Latem, Belgium) and a phylogenetic tree was generated based on the relative abundance proportions of the genera (normalized to 100%), the Pearson’s correlation coefficient and the UPGMA algorithm. Microbiota profiles generated after Illumina or nanopore sequencing were visualized using Microsoft Excel 2010, and ordered based on the sample order in the phylogenetic tree. Alpha-diversity at the genus level was assessed using two metrics: the number of observed genera present with an abundance of at least 1%, and the inverse Simpson index (ISI). Bland-Altman plots were made to explore the comparability of the microbiota profiles generated by Illumina and nanopore sequencing for the six most prevalent genera. These plots show the difference in measured percentages between the two methods versus the mean of the measured percentages.

### 2.8. Sequence Data Availability

The Illumina and nanopore sequence datasets of the nose swab samples, generated and analyzed in the current study, are available in the European Nucleotide Archive (ENA) under accession number PRJEB28612 [29].

## 3. Results

### 3.1. Sample Population

Fifty-one nose swab samples from patients with a respiratory tract infection or sepsis and eight control patients (no infection) were included in the study (Table 1). Most patients were children under the age of 5 years (37/59, 63%). It should be noted that the current analysis was designed to investigate differences between Illumina and nanopore sequencing of nasal microbiota profiles and not to determine possible differences between infection versus no-infection or children versus adult patient populations.

### 3.2. General Sequencing Results

An average of 131,024 raw reads were generated per sample using the Illumina MiSeq platform, with a mean of 91% of raw reads being classified into a mean of 4.4 genera, which were present with an abundance of ≥1% per sample (Table 1). Using nanopore sequencing, an average of 21,907 raw reads were obtained per sample and a mean of 78% of the raw reads were classified into a mean of 4.5 genera, which were present with an abundance of ≥1% per sample (Table 1). The Illumina platform resulted in a significantly higher ISI compared to nanopore; 2.7 vs. 2.2, *p* < 0.0001, paired T. test (Table 1).

For the data generated using nanopore sequencing, 2/59 (3.4%) of the samples were below the cut-off of 500 reads. These samples were excluded from further analysis. Low read numbers ranging from 1–3408 reads for the Illumina platform and 0–56 reads for nanopore were detected in negative control samples (n = 7).

### 3.3. Illumina versus Nanopore Sequencing

Phylogenetic clustering of the taxonomy results (normalized to 100%) generated after Illumina sequencing provided five microbial clades (I–V, Figure 1a). Clade I was dominated by *Moraxella* spp.; II had a mixture of *Moraxella* spp., *Dolosigranulum* sp. and *Corynebacterium* spp.; III *Dolosigranulum* sp. and *Corynebacterium* spp.; IV *Haemophilus* spp.; and V *Staphylococcus* spp. (Figure 1a). When using the Illumina platform, *Corynebacterium* spp., *Moraxella* spp., *Dolosigranulum* sp., and *Streptococcus* spp. were most prevalent, and 1% or more of these genera could be detected in 46, 44, 43, and 32 of the 57 samples analyzed, respectively.

In general, a similar microbiota composition was observed when the genus taxonomy results derived from the two sequencing methods, Illumina and nanopore, were aligned and compared (Figure 1a,b). However, initially, in the nanopore sequenced samples, *Dolosigranulum* sp. was classified in very low abundance (none of that samples had >1%) in the EPI2ME output. By default, the EPI2ME report (EPI2ME version 2.47.537208 and 2.48.690655, Oxford Nanopore Technologies—ONT, Oxford, UK, used May–September 2017) only showed sequence reads for which the num_genus_taxid is 1. The num_genus_taxid represents the total number of different genera out of the top three BLAST classification results. When the num_genus_taxid is 2 or 3, two or three genera are identified in the top 3, respectively, the read is not classified at genus level but at family level (*Carnobacteriaceae* for the genus *Dolosigranulum*), in the EPI2ME report. When we looked at the EPI2ME CSV output file, we noticed that most reads (>95%) with a *Dolosigranulum* genus taxID had a num_genus_taxid of 2. When we added the reads with a num_genus_taxid of 2 to our results (for each genus, dashed lines in Figure 1b), the presence and abundance of *Dolosigranulum* sp. and also *Haemophilus* spp. and *Ornithobacterium* spp. in the nanopore versus the Illumina dataset appeared much more similar (Figure 1a,b).

For nanopore: *Moraxella* spp., *Dolosigranulum* sp. and *Haemophilus* spp. were most prevalent and could be detected with an abundance of at least 1% in 42, 38 and 32 out of 57 samples respectively. Overall, *Moraxella* spp. (33%) were most abundant, followed by *Dolosigranulum* sp. (18%) and *Haemophilus* spp. (18%). To compare the two sequencing platforms, the sum of the percentage of matching genera (sum of agreement) was calculated for each sample (Figure 1c). The highest sum of agreement was 96.9%, the lowest 31.4%, and the median was 69.1%.

To assess the agreement per sample for the six main genera, Bland-Altman plots were generated. With mean differences between 0.9 and −6.0, the detection of *Dolosigranulum* sp., *Moraxella* spp., *Haemophilus* spp., *Staphylococcus* spp., and *Streptococcus* spp. showed good agreement between the two technologies used (Figure 2). However, *Corynebacterium* spp. were detected far more frequently using Illumina sequencing compared to nanopore sequencing (mean difference = 17.1).

To further assess the variability between the Illumina and nanopore sequencing platforms, principal coordinate analysis and PERMANOVA statistics were performed (Appendix A) on the microbiota profiles shown in Figure 1a,b. The platforms only contributed 5.6% to the variations in taxonomy (Illumine versus nanopore), indicating that the platforms perform comparably.

In 2/7 and 6/7 (Illumina and nanopore, respectively) of the negative control samples, bacterial genera were identified (Table 1). Mostly, these genera, which included *Escherichia-Shigella*, *Delphia* and *Pseudomonas* (data not shown), were uncommon in nasal swabs. An exception was negative control C-6 in which 63% of the classified reads, 1500 reads in total, obtained through Illumina sequencing, were identified as *Corynebacterium* spp. In comparison, no reads were generated from the negative control C-6 when using nanopore sequencing.

Compared to the nose swab samples, the number of reads in the negative control samples was maximum 2.7% of the average number or raw reads of 57 samples tested and, therefore, may not have influenced the results obtained from the nasal swabs.

### 3.4. Prevalence of Corynebacterium *spp*.

A striking difference was the significantly lower prevalence and abundance of *Corynebacterium* spp. in the nanopore sequenced samples compared to the samples sequenced by Illumina technology (prevalence based on an abundance of at least 1% per sample: 22/57, 39% vs. 46/57, 81%, *p* < 0.001, Chi squared test; total abundance in the combined nose swab samples: 2.2% vs 19.1%, *p* < 0.001, t-test). There was no obvious explanation for this low prevalence in the EPI2ME CSV files. When we checked whether the ONT 16S rRNA gene primes had a good match with the 16S rRNA gene of *Corynebacterium* spp., using the 16S rRNA gene NCBI database, we found that this was not always the case. *Corynebacterium* spp. that are common residents in the human nose include *C. accolens, C. amycolatum, C. aurimucosum, C. propinquum, C. pseudodiphtheriticum*, and *C. tuberculostearicum* [30,31]. Of these species, both the forward and the reverse primer were not compatible with the 16S rRNA gene of *C. amycolatum*, and there was only an eight basepair stretch (bp 2–9) of the forward primer that annealed to 16S rRNA gene of *C. propinquum*. Thus, the 16S rRNA gene will not be amplified during the PCR using the ONT 16S rRNA gene primers for the *Corynebacterium* species: *C. amycolatum* and *C. propinquum*. Furthermore, the first four bp (5′ end) of the reversed primer could not anneal to the 16S rRNA gene of *C. pseudodiphtheriticum* and *C. tuberculostearicum*. To assess how well the ONT 16S rRNA primers performed in amplifying the 16S rRNA gene, a PCR was done using DNA isolated from pure cultures of five *Corynebacterium* species that we had available in our hospital strain collection (*C. accolens*, *C. amycolatum*, *C. diphtheria*, *C. pseudodiphtheriticum* and *C. striatum*) and four species commonly present in the nasal microbiota (*M. catarrhalis*. *H. influenzae*, *S. aureus,* and *S. pneumoniae*). In agreement with the observed underrepresentation of *Corynebacterium* species in the samples sequenced with the Oxford Nanopore technology, we found that the 16S rRNA gene of the *Corynebacterium* species was poorly amplified (Figure 3).

### 3.5. Re-Basecalling and Analysis of the Nose Swab Samples

To determine whether upgrades in the basecaller and the 16S EPI2ME 16S pipeline improved the detection of genera with an assigned num_genus_taxid of 2, we re-basecalled and re-analyzed the raw reads of all nose swab samples sequenced with the Oxford Nanopore technology. For this, the most recent version of the Guppy basecaller (version 3.2.10, Oxford Nanopore Technologies—ONT, Oxford, UK) and the most recent version of EPI2ME (version 2020.2.10, used April 2020, Oxford Nanopore Technologies—ONT, Oxford, UK) were used.

Instead of the num_genus_taxid, newer versions of the EPI2ME 16S pipeline assign a lowest common ancestor (lca) score of 0 or 1 to the reads in the CSV file. Reads with an lca score of 0 in the newer EPI2ME version are similar to reads with a num_genus_taxid of 1 in the older version, and, by default, are considered to be accurate.

Re-basecalling slightly improved the identification of *Dolosigranulum* sp. (Appendix A). However, still 81% of the reads had an lca score of 1 and were only identified at family level as *Carnobacteriaceae*. No improvement was observed for the identification of *Haemophilus* spp., of which 28% was identified at family level as *Pasteurellaceae* compared 30% in the initial analysis. Based on the highest scoring BLAST identification (top rank), sequence reads that were identified as *Carnobacteriaceae* and *Pasteurellaceae* did belong to the genera *Dolosigranulum* and *Haemophilus*, respectively.

### 3.6. Genus and Species Level Taxonomy on Pure Cultured Single Species Bacteria Using Nanopore Sequencing

To further evaluate how accurately nanopore sequencing of the nasal microbiota performed at genus, and also species level, we sequenced five pure culture bacterial ATCC strains that reflect species that are common to the nasal microbiota. We again followed the development of nanopore data analysis in time and sequenced the ATCC strains twice using flowcell versions R9.2 and R9.4. At genus level, 93.1–99.5% or the sequence reads were accurately identified for 4/5 single species using a R9.2 flowcell and Albacore basecalling. Re-basecalling of the same sequence reads, using Guppy, showed an improvement to 97.0–99.7% accurate identification (Figure 4a). As already observed during sequencing of the nasal microbiota, poor genus identification was found for *H. influenzae* (55.1%, R9.2 flowcell, Albacore, Figure 4a). However, upon re-basecalling using Guppy or re-sequencing using a more recent R9.4 flowcell together with Guppy basecalling, accurate identification of *H. influenzae* at genus level significantly improved to 89.6% in both cases.

At species level, a similar trend of improvement was observed upon re-basecalling sequence reads, generated with a R9.2 flowcell, using Guppy, or using a R9.4 flowcell and Guppy basecalling. An exception was *S. epidemidis*, that, un-expectantly, showed poorer identification with the R9.4- compared to the R9.2 flowcell, with 58.9% of the sequence reads being mis-identified as *S. saccharolyticus* (Figure 4b).

## 4. Discussion

In this study, we compared and evaluated two 16S ribosomal gene sequencing strategies based on Illumina and nanopore technologies by analyzing the nasal microbiota composition of 59 human nose swab samples. In general, both sequencing techniques performed comparably at genus level except for the detection of *Corynebacterium* spp., a main and established genus in the nasal microbiota that was poorly detected by the Oxford Nanopore platform. New releases of a basecaller and of the nanopore flowcell led to improved genus and species identification but not for all species tested.

Upon comparing Illumina versus nanopore sequencing of the nasal microbiota samples tested, a comparable average diversity of 4.4 and 4.5 bacterial genera (Illumina versus nanopore) was detected per sample. The ISI—a measure of diversity that takes the number as well as the relative abundance of species in an environment into account—indicated greater bacterial genus diversity when Illumina sequencing was compared to nanopore, on average 2.7 versus 2.2 respectively. These numbers are lower than a previously published ISI of 4.1 for the nasal microbiota [30]. This difference may have been the result of the fact that we calculated our values based on genera instead of using operational taxonomic units (OTUs), which are more diverse and normally used for Illumina sequencing. The relative young age of the individuals sampled in the current study and the fact that many were sampled during active infection may also have resulted in our relatively low ISI values [32].

The most dominant genera detected by the Illumina platform were: *Corynebacterium*, *Dolosigranulum*, *Haemophilus, Moraxella, Staphylococcus*, and *Streptococcus.* Previous culture- and next generation sequence approaches have revealed that these are well established genera in the nasal microbiota [31].

Initially, most of the nanopore sequenced reads derived from bacteria with the genus *Dolosigranulum* were identified at family level only i.e., *Carnobacteriaceae*, which appeared to be due to fixed cut-off restrictions in the output of the Oxford Nanopore Technologies EPI2ME 16S workflow. 

In the EPI2ME 16S workflow, basecalled nanopore sequence reads are blasted against the NCBI 16S rRNA gene database. Although it is possible that certain species are not represented in the NCBI database, this was not the case for *Dolosigranulum* sp. as 16S rRNA gene sequences of at least two strains are present (taxid 29394 and 883103). However, exactly because there were only two 16S rRNA gene sequences of *Dolosigranulum* sp. present in the NCBI database, the condition of a top three blast hit with similar genera (num_genus_taxid is 1, or lca is 0), which is a requirement for reads to be classified using the EPI2ME 16S workflow, cannot be met. Thus, the limited number of two *Dolosigranulum* 16S rRNA genes in the NCBI 16S rRNA gene database is probably why the EPI2ME workflow failed to identify this genus. Besides *Dolosigranulum* sp., the bacterial genera *Haemophilus* and *Ornithobacterium* were also identified more abundantly when read with a top three blast hit with two similar genera (num_genus_taxid is 2) together with reads with a top three blast hit with three similar genera (num_genus_taxid is 1) were included in the analysis. It did not become clear to us why this was the case.

When taking into account the inclusion of sequence reads with a num_genus_ taxid of 1 or 2, comparison of the two sequencing platforms resulted in a median sum of agreement of 69.1%, with the main genera *Dolosigranulum, Moraxella*, *Haemophilus*, *Staphylococcus,* and *Streptococcus* showing good agreement. *Corynebacterium*, however, was severely underrepresented in the taxonomy data generated after analysis of the nanopore sequencing results, even when reads with a num_genus_taxid other than 1 were included. Blast analysis established that two *Corynebacterium* species, *C. amycolatum* and *C. propinquum,* known to be habitants of the nasal microbiota [31], could not be detected due to potential incompatibility of the nanopore 16S rRNA gene sequence primers. In agreement with this analysis, we observed that the 16S rRNA gene of several pure culture *Corynebacterium* spp. could not be amplified using the nanopore 16S primes. Incomplete annealing at the first four 5′ base pairs of the nanopore reverse primers, applicable for *C. pseudodiphtheriticum* and *C. tuberculostearicum*, may additionally have resulted in a low prevalence of *Corynebacterium* species. However, the first four 5′ base pairs of this reverse primer also did not match several other species that were detected in high abundance (including *M. catarrhalis* and *M. nonliquefaciens*), which tends to negate the hypothesis that poor annealing of the nanopore reverse primer led to an underrepresentation of *C. pseudodiphtheriticum* and *C. tuberculostearicum*. Still, mismatching of the nanopore primers and poor amplification of the 16S rRNA gene of another bacterial genus, *Bifidobacterium*, has been reported [33]. A PCR bias due to the relatively high genomic GC-content may be another explanation why the genus *Corynebacterium* was underrepresented in the samples sequenced using Oxford Nanopore technology [34]. With respect to nasal microbiota profiling, our results indicate that researchers should take into account the fact that different sequencing platforms and pipelines may generate different results. However, it is usually (due to cost) not feasible to perform research microbiota profiling using multiple sequencing platforms. It should also be noted that Illumina and nanopore sequencing technologies are constantly evolving and improvements in available sequencing hardware and software platforms are constantly being made.

In this respect, we also compared taxonomic analysis performance using pure cultured bacterial isolates and the newest ONT hardware and sequence basecalling platform (R9.4 flowcells and Guppy). At genus level, we found that at least 93% of the reads were accurately identified for 4/5 ATCC strains tested with a R9.2 flowcell, and an improvement for the remaining strain when we used Guppy instead of Albacore basecalling software or a R9.4 compared to a R9.2 flowcell.

Bacterial taxonomic identification at species level can be of clinical importance, as it can help guide antibiotic prescription in cases of infection, or potentially identify (prophylactic) species that suppress nasal colonization of opportunistic pathogens. For example, previous studies have demonstrated that *S. epidermidis* may secrete a serine protease (Esp) that is able to inhibit nasal colonization by *S. aureus* [35]. Further, *S. mitis* has been negatively associated with nasal colonization by methicillin-resistant *S. aureus* (MRSA)—apparently being able to inhibit the growth of MRSA by a hydrogen peroxide-mediated mechanism [36].

When we addressed species level identification of nanopore sequence reads, we found that 4/5 pure culture species were accurately identified when using a R9.4 flowcell and Guppy basecalling. However, species identification of *S. epidermidus* was found to occur with almost 60% of reads being mis-classified as *S. saccharolyticus*. This mis-classification may have been the result of a high degree of sequence similarity between the *S. epidemidis* and *S. saccharolyticus* 16S rRNA gene. Because the bacteria were grown under aerobic conditions in which anaerobic *S. saccharolyticus* does not grow, contamination of the *S. epidermidus* culture with *S. saccharolyticus* before DNA isolation is not plausible.

## 5. Conclusions

In conclusion, the current study shows that microbiota profiling of the human nasal microbiota, using nanopore sequencing platform, is comparable to Illumina sequencing at the genus level and above. However, nanopore sequencing may not accurately identify bacteria within the genus *Corynebacterium*. At the species level, it appears that advances still need to be made to improve the accuracy of taxonomic classification by nanopore sequencing (as with other sequencing technologies). Since our initial comparative studies began, accurate taxonomic assignment at species level using nanopore sequencing continues to improve, with advances in reducing the relatively high error rate of nanopore sequencing, generating obvious advantages. Such changes are to be welcomed. However, constantly evolving hardware and software outputs complicate downstream data analysis and make the comparison of historically published results with more recent results potentially problematic.

## Figures and Tables

**Figure 1 genes-11-01105-f001:**
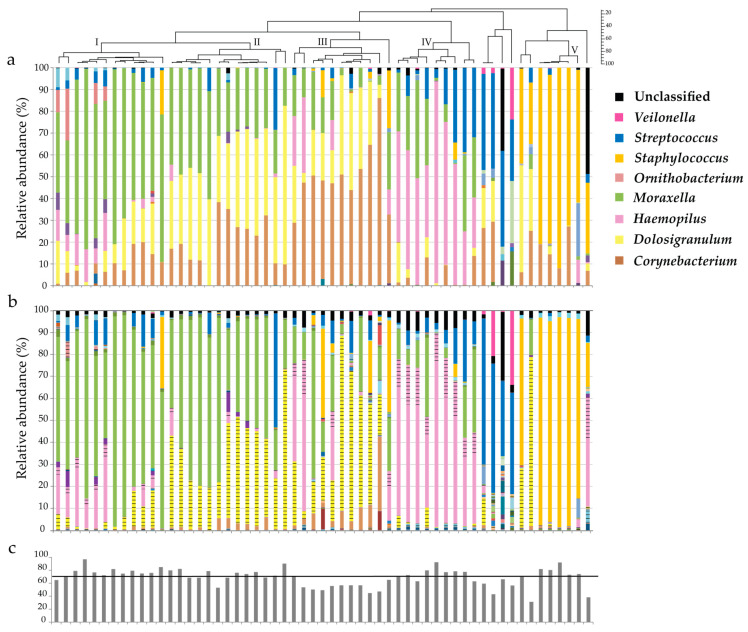
Nasal microbiota profiles generated using nanopore and Illumina 16S rRNA gene sequencing. DNA was isolated from 57 nose swab samples, and 16S rRNA gene sequencing was performed using both Illumina (**a**) and nanopore (**b**) technologies. Each bar in the graph represents a nasal microbiota profile from a single individual. The dashed lines in (**b**) represent genera that, by default, were reported as unclassified at genus level in the EPI2ME report but were identified when next to reads with a top three blast hit with one genera (num_genus_taxid is 1); reads with a top three blast hit with two genera (num_genus_taxid is 2) were also included. A phylogenetic tree was generated by Pearson/UPGMA clustering of bacterial genera in microbiota profiles, as determined using Illumina sequencing. To compare between the two techniques, the sample order of the samples that were sequenced with the Oxford Nanopore platform was matched to the sample order of the samples that were sequenced with the Illumina platform, and the percentage of agreement was calculated for each nose swab sample (**c**). The horizontal black line in (**c**) indicates the mean percentage of agreement.

**Figure 2 genes-11-01105-f002:**
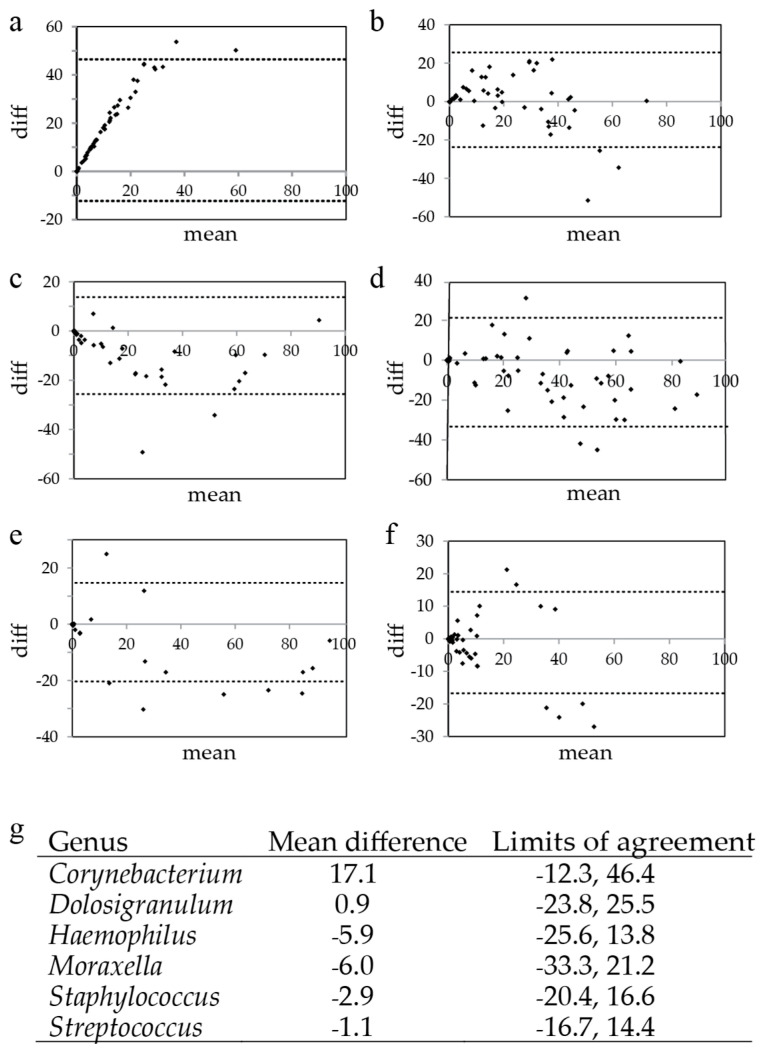
Bland–Altman plots of six main genera present in the nasal microbiota. Bland–Altman plots were generated for the six main genera: (**a**) *Corynebacterium*, (**b**) *Dolosigranulum*, (**c**) *Haemophilus*, (**d**) *Moraxella,* (**e**) *Staphylococcus*, and (**f**) *Streptococcus*. For each genus, the mean difference between the two sequence methods (Illumina versus nanopore) and the limits of agreement (95% reference interval) were calculated and shown (**g**).

**Figure 3 genes-11-01105-f003:**
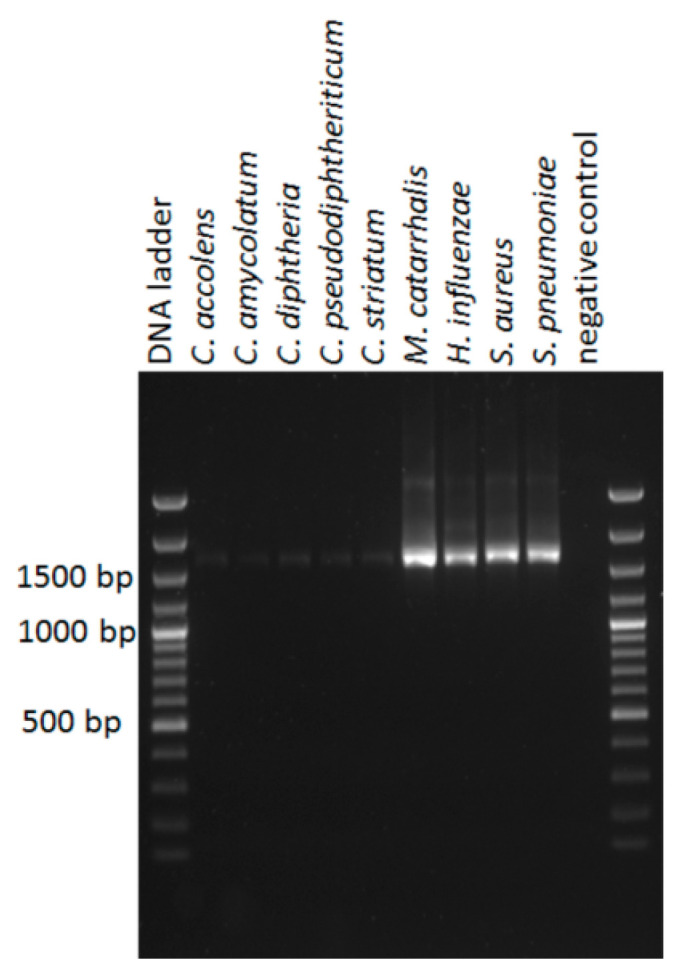
Agarose gel with 16S rRNA gene amplicons. Total DNA was isolated from pure bacterial cultures in a similar manner as the isolation of DNA from the nasal swab samples; the DNA concentration was determined by picogreen and a PCR was performed as described for nanopore sequencing using equal amounts of template DNA, with the exception that 30 PCR cycli instead of 25 cycli were used.

**Figure 4 genes-11-01105-f004:**
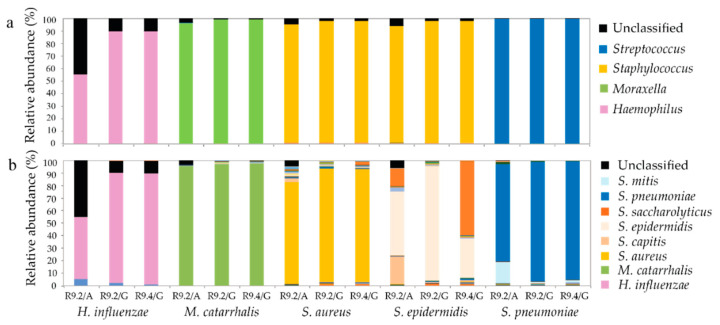
Genus and species level identification on pure culture species. Pure cultures of bacterial ATCC strains were sequenced using an R9.2 or R9.4 nanopore flowcell and Albacore or Guppy basecalling. Taxonomic assignment was performed at genus (**a**) and species (**b**) level using the EPI2ME 16S pipeline and the following thresholds: read length ≥1400 bp ≤ 1700 bp, num_genus_taxid is 1 or lca is 0 and accuracy ≥80%, QC ≥ 7 when albacore basecalling was used, or accuracy ≥85%, QC score ≥9 when Guppy basecalling was used. Similar criteria and the highest scoring BLAST identification (top rank) was used for species level identification. A is Albacore; G is Guppy basecalling.

**Table 1 genes-11-01105-t001:** Nose swab samples of individuals and negative controls that were sequenced using and Illumina and nanopore 16S rRNA gene sequencing technologies. ^(a)^ = a maximum of 5000 raw Illumina sequence reads were analyzed for the classification of genera. ^(b)^ = samples with read numbers below the 500 read cut-off. NA = not applicable.

Sample Information				Illumina Technology				Nanopore Technology			
Sample	Infection	Age (Years)	16S Copies	Raw Reads	Percentage Reads Classified (%) ^(a)^	Genera Identified ≥1%	Genera (ISI)	Raw Reads	Percentage Reads Classified (%)	Genera Identified ≥1%	Genera (ISI)
1	yes	3.50	5 × 10^5^	133,880	92	5	4.2	34,944	77	5	3.0
2	yes	0.92	1 × 10^5^	186,250	95	5	1.9	15,254	79	3	2.2
3	yes	2.00	3 × 10^5^	1661	94	5	4.1	39,474	77	4	2.9
4	yes	1.50	3 × 10^5^	154,877	96	7	4.6	36,608	76	6	2.3
5	yes	9.00	3 × 10^5^	114,702	97	5	3.5	5107	59	4	2.7
6	yes	2.00	3 × 10^5^	22,805	97	5	2.7	31,642	52	4	1.7
7	yes	5.00	2 × 10^5^	1940	88	8	3.8	2246	57	6	3.1
8	yes	4.00	3 × 10^5^	24,214	100	4	1.2	10,174	62	3	1.2
9	yes	1.67	4 × 10^5^	104,134	93	9	2.5	21,462	68	6	2.6
10	yes	8.00	2 × 10^5^	186,945	96	3	2.5	923	68	2	1.6
11	yes	11.00	2 × 10^5^	120,867	95	3	3.0	27,569	78	3	1.6
12	yes	0.42	4 × 10^5^	25,743	98	3	3.0	5127	66	3	2.2
13	yes	15.00	4 × 10^5^	261,123	95	4	2.7	12,572	66	5	2.0
14	yes	2.17	1 × 10^5^	6246	97	4	3.0	20,441	89	3	2.7
15	yes	3.80	3 × 10^5^	68,095	91	3	2.3	27,077	90	4	2.5
16	yes	2.40	1 × 10^5^	119,295	84	7	2.9	2978	85	6	2.6
17	yes	0.80	2 × 10^5^	74,902	96	3	1.5	4408	91	2	1.1
18	yes	61.00	3 × 10^3^	77,851	86	6	3.4	2141	82	8	4.1
19	yes	0.90	3 × 10^5^	74,730	85	4	2.3	20,584	82	6	1.6
20	yes	0.80	3 × 10^5^	113,078	93	3	2.4	10,974	91	3	1.9
21	yes	78.00	2 × 10^5^	131,837	90	2	1.7	21,449	93	1	1.0
22	yes	1.70	3 × 10^5^	162,890	85	4	2.4	23,530	92	5	1.8
23	yes	2.30	2 × 10^5^	83,596	92	8	4.4	15,748	88	7	3.2
24	yes	73.00	2 × 10^5^	83,947	84	4	2.0	3181	88	5	3.3
25	yes	2.60	5 × 10^5^	28,221	92	3	3.0	15,453	50	3	3.1
26	yes	65.00	3 × 10^5^	77,012	82	7	4.5	31,461	85	6	2.8
27	yes	0.80	1 × 10^5^	58,962	85	3	2.5	23,652	90	3	1.5
28	yes	3.00	5 × 10^5^	57,600	86	6	3.7	22,991	84	7	3.4
29	yes	57.00	2 × 10^5^	129,131	94	2	1.5	48,167	90	1	1.1
30	yes	0.40	6 × 10^5^	180,796	88	3	2.9	3997	65	4	2.3
31	yes	0.90	4 × 10^5^	547,695	98	4	2.7	15,626	80	7	1.7
32	yes	23.00	8 × 10^5^	750,669	97	3	1.8	6653	67	2	1.6
33	yes	3.40	1 × 10^5^	924,890	98	7	3.3	25,148	74	7	2.1
34	yes	4.10	1 × 10^6^	31,896	94	5	4.0	15,979	49	4	2.7
35	yes	14.00	3 × 10^5^	79,970	90	3	2.1	40,551	88	3	1.4
36	yes	0.10	3 × 10^5^	113,047	88	3	1.7	50	76	NA	NA
37 ^(b)^	yes	0.40	3 × 10^5^	59,397	88	4	2.9	51,254	63	11	3.6
38	yes	0.30	6 × 10^5^	7421	99	3	1.4	41,757	89	2	1.4
39	yes	1.10	3 × 10^5^	121,819	86	3	2.6	6340	86	6	1.9
40	yes	0.20	2 × 10^5^	83,457	83	4	2.4	59,923	82	6	1.9
41	yes	4.20	4 × 10^5^	92,006	87	4	2.9	17,785	90	4	2.3
42	yes	0.10	1 × 10^5^	36,248	90	4	2.0	45,047	92	3	1.9
43	yes	0.10	2 × 10^5^	55,585	92	5	2.3	47,084	92	3	1.4
44	yes	0.40	3 × 10^5^	101,465	87	5	2.7	5288	80	6	1.6
45	yes	1.70	7 × 10^5^	92,476	89	3	1.9	49,104	55	2	1.1
46	yes	0.50	3 × 10^5^	72,068	88	4	2.1	50,486	80	6	1.5
47	yes	0.10	5 × 10^5^	90,128	80	6	4.0	107,161	91	6	3.2
48 ^(b)^	yes	67.00	2 × 10^5^	51,826	94	5	1.3	8	75	NA	NA
49	yes	0.30	9 × 10^5^	1148	82	8	4.3	14,673	66	3	1.5
50	yes	3.30	5 × 10^6^	39,030	83	3	2.6	12,239	66	3	2.0
51	yes	56.00	5 × 10^6^	2191	85	7	3.3	17,248	64	7	2.4
52	no	28.00	3 × 10^5^	193,859	96	2	1.2	6,789	91	1	1.0
53	no	62.00	2 × 10^5^	262,184	89	3	2.3	18,680	88	2	1.8
54	no	8.10	2 × 10^5^	308,123	83	5	2.8	13,741	89	4	2.3
55	no	7.20	3 × 10^5^	203,242	100	6	3.4	15,490	84	6	4.3
56	no	14.90	1 × 10^5^	235,820	92	3	1.4	18,318	88	8	5.0
57	no	5.40	9 × 10^5^	90,422	86	3	2.8	11,207	87	4	2.2
58	no	7.10	6 × 10^5^	103,176	87	5	2.9	19,604	73	7	2.7
59	no	6.40	1 × 10^5^	111,844	93	4	1.6	17,971	88	3	1.1
Average	**NA**	**12.5**	**8** **× 10^5^**	**131,024**	**91**	**4.4**	**2.7**	**21,907**	**78**	**4.5**	**2.2**
Control											
C-1	NA	NA	<1 × 10^2^	6	0	0	NA	7	57	4	NA
C-2	NA	NA	<1 × 10^2^	1	0	0	0	42	74	8	NA
C-3	NA	NA	<1 × 10^2^	1	0	0	0	33	42	9	NA
C-4	NA	NA	<1 × 10^2^	3	0	0	0	35	51	11	NA
C-5	NA	NA	<1 × 10^2^	2	0	0	0	15	67	3	NA
C-6	NA	NA	2 × 10^2^	2440	98	4	4	56	91	6	NA
C-7	NA	NA	3 × 10^2^	3408	94	18	18	0	0	0	NA

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
