# Peer review of "Comparison of Illumina versus Nanopore 16S rRNA Gene Sequencing of the Human Nasal Microbiota"

_genes, 2020, doi:10.3390/genes11091105_

Round 1
Reviewer 1 Report
Heikema reports a comparison between two sequencing techniques, Illumina and Nanopore, for 16S taxonomic identification of mucosal microbiota in nasal swabs. The article is well written and concise with comparison data presented adequately. Outside of a few formatting errors the article should be accepted as is with the following recommendation. The title of the article is a comparison between the two techniques, however in the abstract, Heikema reports only the Nanopore results. Please make the last few lines of the abstract more in line with the conclusion.Author Response
We thank the reviewer for the critical reading of our manuscript and for giving constructive suggestions for improvement. We have made changes to the manuscript accordingly and we hope that the revised manuscript satisfactorily addressed the concerns of the reviewer. Below, we respond to the reviewer comments point by point.
Comments from Reviewer:
Reviewer reports:
Reviewer 1.
The title of the article is a comparison between the two techniques, however, in the abstract, Heikema reports only the Nanopore results. Please make the last few lines of the abstract more in line with the conclusion.
Page 1, lines 6-11
Thank you for this suggestion. We added the following to the abstract.
“In conclusion, the current study shows that the nanopore sequencing platform is comparable with the Illumina platform in detection bacterial genera of the nasal microbiota, but the nanopore platform does have problems in detecting bacteria within the genus Corynebacterium. At the species level, thorough validation is recommendable, and advances still need to be made to improve the accuracy of taxonomic classification by nanopore sequencing.
Reviewer 2 Report
In their study „Comparison of Illumina versus nanopore 16S rRNA gene sequencing of the human nasal microbiota” Heikema et al. sequenced 16S rRNA 95 nasal swabs using the MiSeq from Illumina and the Oxford nanopore sequencer to determine the bacterial composition. Overall, the manuscript is of importance regarding the comparison of the two sequencing types and is well written.
I have only minor remarks/questions:
Page 5, line 11: A space is missing between the bracket and basecalling.
Page 13, lines 21-43: Do you know why there is such an incompatibility of the nanopore primers with 16S rRNA from Corynebacterium species? It is quite surprising, because many Corynebacterium strains are used a model organisms. Is this problem also the case in other species? Did you find any information about this in the literature? Is it for example possible to use custom-made primers for sequencing, if such a problem occurs?
Page 14, line 11-12: Move the paragraph starting with because in line 12 to line 11.
Author Response
We thank the reviewer for the critical reading of our manuscript and for giving constructive suggestions for improvement. We have made changes to the manuscript accordingly and we hope that the revised manuscript satisfactorily addresses the concerns of the reviewer. Below, we respond to the reviewer comments point by point.
Comments from Reviewers:
Reviewer 2.
Page 5, line 11: A space is missing between the bracket and basecalling.
Thank you for this comment. We could not find this issue on page 5 line 11 in our version of the manuscript but we added a space on page 6 line 3 and 7 between the bracket and basecalling.
Page 13, lines 21-43: Do you know why there is such an incompatibility of the nanopore primers with 16S rRNA from Corynebacterium species? It is quite surprising, because many Corynebacterium strains are used a model organisms.
Thank you for your comment. Some time ago we contacted ONT to inquire about the primer mismatch for Corynebacterium species. They mentioned that the primers were designed as optimal as possible but that it is was possible that there was primer incompatibility due to species to species variation in the 16S rRNA gene region the primers are directed to.
Is this problem also the case in other species? Did you find any information about this in the literature?
There is a publication that describes primer mismatching and poor amplification of the Bifidobacterium 16S rRNA gene (doi: 10.1002/2211-5463.12590). We included this reference in the manuscript together with the following text.
Page 17, lines 3-5: “Still, mismatching of the nanopore primers and poor amplification of the 16S rRNA gene of another bacterial genus, Bifidobacterium, has been reported [27].”
Is it for example possible to use custom-made primers for sequencing, if such a problem occurs?
The (presumably patented) chemistry needed to guide the DNA to the pore of the nanopore flowcell is imbedded in the nanopore primers which are part of an ONT 16S library kit. It surely is desirable to have custom-made primers, optimized for a specific microbiota, but this option is not available yet at ONT.
Page 14, line 11-12: Move the paragraph starting with because in line 12 to line 11.
Page 17, line 35-38: We have moved the paragraph as suggested.
Reviewer 3 Report
- This study amplified partial 16S rRNA gene sequences through Illumina and Nanopore sequencing platforms and it is not clear how authors were able to annotate at species-level? I think restraining the annotations to genera-level is meaningful as the homology search provided best hits and one can find several species-level annotations with the same identity and coverage, which is why with partial 16S sequence cannot provide species-level resolution.
- Why were different 16S variable regions were chosen for Illumina and Nanonopore? Authors should consider compositional bias caused by primers and V-regions.
- Beta-diversity analysis should be presented along with stats e.g. PERMANOVA to show Inter-sample variability between the two sequencing approaches, without this, I'm not sure how to interpret the microbiota profiles in Figure 1.
- A uniform font should be maintained throughout the manuscript.
Author Response
We thank the reviewer for the critical reading of our manuscript and for giving constructive suggestions for improvement. We have made changes to the manuscript accordingly and we hope that the revised manuscript satisfactorily addressed the concerns of the reviewer. Below, we respond to the reviewer comments point by point.
Comments from Reviewer:
Reviewer 3.
1. This study amplified partial 16S rRNA gene sequences through Illumina and Nanopore sequencing platforms and it is not clear how authors were able to annotate at species-level? I think restraining the annotations to genera-level is meaningful as the homology search provided best hits and one can find several species-level annotations with the same identity and coverage, which is why with partial 16S sequence cannot provide species-level resolution.
Thank you for this comment, we totally agree.
2. Why were different 16S variable regions were chosen for Illumina and Nanonopore? Authors should consider compositional bias caused by primers and V-regions.
With the nanopore sequencing method, the complete 16S gene is sequenced using commercial primers that are supplied by ONT. These are the only primers available for 16S rRNA sequencing using this platform. It, therefore, is not possible to sequence the same region as was done with the Illumina platform. We agree with the reviewer that bais can be caused by the primers.
3. Beta-diversity analysis should be presented along with stats e.g. PERMANOVA to show Inter-sample variability between the two sequencing approaches, without this, I'm not sure how to interpret the microbiota profiles in Figure 1
Thank you for this valuable suggestion. We added a PCoA analysis to the manuscript together with PERMANOVA stats.
The following was added to the text.
Page 2 line 3-7
"To further assess the variability between the Illumina and nanopore sequencing platforms, principal component analysis, and PERMANOVA statistics were performed (Figure S2) on the microbiota profiles shown in figure 1a and figure 1b. The platforms only contributed 5.6% to the variations in taxonomy (Illumine versus nanopore), indicating that the platforms perform comparably".
Also, a supplemental figure was added, Figure S2, see the revised manuscript.
4. A uniform font should be maintained throughout the manuscript.
We adjusted the font and the font size throughout the manuscript including the figures.